# Theoretical Investigation of Delafossite-Cu_2_ZnSnO_4_ as a Promising Photovoltaic Absorber

**DOI:** 10.3390/nano13243111

**Published:** 2023-12-10

**Authors:** Seoung-Hun Kang, Myeongjun Kang, Sang Woon Hwang, Sinchul Yeom, Mina Yoon, Jong Mok Ok, Sangmoon Yoon

**Affiliations:** 1Materials Science and Technology Division, Oak Ridge National Laboratory, Oak Ridge, TN 37831, USA; kangs@ornl.gov (S.-H.K.); yeoms@ornl.gov (S.Y.);; 2Department of Physics, Pusan National University, Busan 46241, Republic of Korea; maudwnsm@gmail.com; 3Department of Physics, Gachon University, Seongnam 13120, Republic of Korea; criori2010@gachon.ac.kr

**Keywords:** computational material design, photovoltaic absorber, delafossite oxides

## Abstract

In the quest for efficient and cost-effective photovoltaic absorber materials beyond silicon, considerable attention has been directed toward exploring alternatives. One such material, zincblende-derived Cu2ZnSnS4 (CZTS), has shown promise due to its ideal band gap size and high absorption coefficient. However, challenges such as structural defects and secondary phase formation have hindered its development. In this study, we examine the potential of another compound, Cu2ZnSnO4 (CZTO), with a similar composition to CZTS as a promising alternative. Employing ab initio density function theory (DFT) calculations in combination with an evolutionary structure prediction algorithm, we identify that the crystalline phase of delafossite structure is the most stable among the 900 (meta)stable CZTO. Its thermodynamic stability at room temperature is also confirmed by the molecular dynamics study. Excitingly, this new phase of CZTO displays a direct band gap where the dipole-allowed transition occurs, making it a strong candidate for efficient light absorptions. Furthermore, the estimation of spectroscopic limited maximum efficiency (SLME) directly demonstrates the high potential of delafossite-CZTO as a photovoltaic absorber. Our numerical results suggest that delafossite-CZTO holds promise for future photovoltaic applications.

## 1. Introduction

The efficiency of a photovoltaic cell depends heavily on the material used as an absorber. Silicon (Si) is currently the most widely used absorber due to its abundance and affordability. Though the size of its band gap is favorable for visible light absorption, it exhibits limited absorption properties in the visible spectrum owing to the indirect nature of its band gap. Accordingly, Si requires thick wafers to absorb light, leading to low power conversion efficiency (PCE) and increasing costs [1]. GaAs [2], Cu_2_InGaSe_4_ (CIGS) [3], and halide perovskites such as CH_3_NH_3_PbI_3_ [4] have been explored as alternative materials to overcome these limitations. GaAs and CIGS have better absorption properties than Si but are made of expensive elements such as gallium (Ga) and indium (In) [5,6]. Halide perovskites are emerging as a promising class of photovoltaic absorbers due to their outstanding absorption properties [4]. However, they currently suffer from various instabilities associated with organic molecules. Finding new materials with superior absorption properties, high stability, and lower production prices is still a key obstacle for developing next-generation photovoltaic cells and making solar energy an alternative energy resource to fossil fuels [7,8].

Cu_2_ZnSnS_4_ (CZTS), which possesses a zincblende-derived structure, has attracted considerable attention as a potential alternative to CIGS, primarily due to its potential to overcome the limitation of CIGS [9,10]. CZTS comprises earth-abundant, non-toxic, and cost-effective elements, namely zinc (Zn) and tin (Sn), and exhibits favorable properties such as a band gap size of 1.45 eV [11] and a high absorption coefficient of 10^4^ cm^−1^ [12]. However, despite extensive research, CZTS-based solar cells have yet to surpass an efficiency of 12% [13], largely owing to issues such as phase separation and the emergence of several structural defects [14]. On the other hand, Cu_2_ZnSnO_4_ (CZTO), which consists of oxygen (O) with the same number of valence electrons as other chalcogen elements, could potentially demonstrate similar physical properties to CIGS and CZTS. Furthermore, if successfully synthesized, CZTO would offer its own advantages, as oxide compounds generally exhibit high stability under various ambient environments [15]. Nonetheless, it is worth noting that, to the best of our knowledge, the thermodynamic and physical properties of a quaternary oxide CZTO have not been investigated yet in both theory and experiment.

In this paper, we explore the most stable crystal structure within the quaternary CZTO compound, known as delafossite-CZTO. Delafossite oxide belongs to a class of metal oxides where triangular A and hexagonal BO_2_ atomic layers stack alternately, forming a unique three-dimensional structure. This stable phase is identified through the particle swarm optimization (PSO) process. The delafossite-CZTO exhibits a direct band gap and dipole-allowed transitions, facilitating robust light absorption. Spectroscopic limited maximum efficiency (SLME) analysis indicates that this new phase of CZTO offers higher efficiency compared to other oxide materials. This work highlights the potential of delafossite-CZTO as an efficient and promising photovoltaic absorber with desirable properties such as high efficiency, low cost, low toxicity, and high stability.

## 2. Results

By employing the particle swarm optimization (PSO) algorithm based on ab initio density functional theory (DFT) calculations, we have identified the thermodynamically stable crystal structures among quaternary CZTO compounds. During the optimization process, we relaxed the volume of the crystal structure while keeping the ratio of atoms fixed (i.e., Cu:Zn:Sn:O = 2:1:1:4). Among 900 crystal structures considered, those deviating by 0.7 meV/atom or less from the most stable structure are presented (denoted by a blue circle in Figure 1a). The most stable structure, resembling a delafossite oxide, features alternating triangular and hexagonal layers of Cu and Zn_0.5_Sn_0.5_O_2_ (denoted as delafossite-CZTO) (see Figure 1c). Surprisingly, the expected stable kesterite and stannite phases, akin to CIGS and CZTS, were energetically less favorable. The total energy of kesterite and stannite structures was 200 and 204 meV/atom higher, respectively, than delafossite-CZTO. Refer to Appendix B for computational and experimental details.

When determining the stability of a structure, it is not enough to simply rely on the strength of its cohesive energy. A structure could still be prone to spontaneous changes if there are negative frequencies in its vibrations, which would suggest structural instability. To assess this, we studied the vibrational properties of delafossite-CZTO, as shown in Figure 2a. The absence of negative frequencies along the high-symmetry line in the phonon band suggests that delafossite-CZTO is dynamically stable. However, while phonon dispersion can provide some insight into the structure’s stability, it does not definitively tell us whether the structure might collapse at a specific temperature. To determine whether the new delafossite-CZTO structure can remain stable at room temperature (300 K), we used canonical molecular dynamics (MD) simulations using a 3 × 3 × 3 supercell. Figure 2b shows how the total potential energy evolved over a 3 ps MD simulation and provides a snapshot of the final structure. Our findings confirm that delafossite-CZTO structure maintains stability at room temperature (300 K).

Having checked the structural stabilities of delafossite-CZTO, we move to the electronic structure and its properties. The efficiency of a photovoltaic absorber is determined by several factors, including the bulk optical property of the absorber, the type and distribution of structural defects within the absorber, and the artifacts induced by the fabrication of the device. Among these, the intrinsic factors, i.e., the bulk optical properties, are mainly determined by the electronic structure of the absorber material. Thus, we calculated the band structure of delafossite-CZTO, which emerges as the most stable phase, using DFT calculations at the GGA-PBE and GW_0_ levels. As shown in Figure 3a, delafossite-CZTO exhibits a direct band gap at the Γ point in both GGA-PBE and GW_0_ calculations, suggesting that the direct band transition positions in lower energy than other indirect band transitions. Here, the valence and conduction bands in delafossite-CZTO comprise Cu d and Zn/Sn p orbitals. The transition between the two orbitals is dipole-allowed across the direct band gap. Thus, the light is absorbed strongly when the dipole-allowed transition occurs [16]. The size of the direct band gap is estimated to be 1.58 eV (GW_0_), very close to that of the ideal photovoltaic absorber (~1.34 eV) according to the Shockley and Queisser (SQ) criterion [1]. These results indicate that delafossite-CZTO has a preferable electronic structure for the application of a photovoltaic absorber. To note, the gap size differs significantly by 1.32 eV for 1.58 eV (GW_0_) and 0.26 eV (GGA-PBE). The GGA-PBE method commonly underestimates band gaps due to self-interaction errors. The GW_0_ approach corrects this error, providing a more accurate band gap prediction.

The absorption spectrum ε_2_(ω) (the imaginary part of the dielectric function) of delafossite-CZTO is estimated based on the GW calculations, as shown in Figure 3b. We further solve the Bethe–Salpeter equation (BSE) with quasiparticle energy bands to consider the photo-excited electron and hole interactions. We chose the twenty occupied orbitals, and six unoccupied orbitals suffice to converge ε_2_(ω) up to 5.0 eV. The absorption spectra obtained from GW + BSE are compared with the non-interacting case (GW + RPA). Both GW + BSE and GW + RPA spectra do not show superior optical absorption in energy below E_g_ (denoted as the dashed line in Figure 3b), but the electron–hole interactions shift the overall absorption spectra to the lower photon energy range [10,17]. The red shift of the absorption spectra above E_g_ generally improve the efficiency of a photovoltaic absorber because the light absorption within the solar spectral range is enhanced by the redshift. In addition, an exciton forms at 2.08 eV (indicated by the red line in Figure 3b), further augmenting the absorption coefficient within the visible range (see Figure 4a).

To computationally screen new absorber candidates effectively, having a descriptor that captures a photovoltaic absorber’s intrinsic properties is essential. The classical descriptor is simply the band gap, which Shockley and Queisser (SQ) suggested [18]. Based on this SQ criterion, the optimal band gap is ~1.34 eV for the maximum solar conversion efficiency to 33.16% for a single-junction solar cell. However, it is obvious that this descriptor is insufficient because the numerous materials with similar sizes of the optimized band gap exhibit poor photovoltaic efficiency in experiments. A more sophisticated model, a widely used descriptor of the efficiency of a photovoltaic absorber, is the spectroscopic limited maximum efficiency (SLME; η) [19]. This can be estimated by DFT calculations as follows:$$\eta =\frac{P_{max}}{P_{in}}=\frac{{max\left\{\left(j_{sc}-j_{0}(e^{eV/kT}-1)\right)V\right\}}_{V}\;}{\int_{0}^{\infty }EI_{SUN}\left(E\right)dE}$$

Here, $j_{sc}=e\int_{0}^{\infty }a\left(E\right)I_{SUN}\left(E\right)dE$ is the short-circuit current density where $a\left(E\right)=1-e^{-2\alpha \left(E\right)L}$ is the photon absorptivity and $I_{SUN}\left(E\right)$ is the AM1.5G solar spectrum. $j_{0}=j_{0}^{r}/f_{r}$ is the reverse saturation current density, $j_{0}^{r}=e\pi \int_{0}^{\infty }a\left(E\right)I_{bb}\left(E,T\right)dE$ is the rates of emission and absorption through the cell surface, $I_{bb}\left(E,T\right)$ is the black-body spectrum at temperature T, $f_{r}=e^{(E_{g}-E_{g}^{da})/kT}$ is the fraction of the radiative recombination current, and $E_{g}$ and $E_{g}^{da}$ denote the band gap and direct-allowed band gap. Accordingly, the property-related inputs are the absorption coefficient α(E), the fraction of the radiative recombination current $f_{r}$, the thickness *L*, and the temperature *T*. Note that extrinsic effects such as structural defects and fabrication-induced artifacts are not considered in the SLME.

The SLME of delafossite-CZTO is calculated as a function of thickness to evaluate the potential efficiency of this new material, as shown in Figure 4b. For the thickness *L* = 2.0 μm and the temperature *T* = 300 K, the SLME of delafossite-CZTO is about 28.2%. The SLME of CZTO is relatively high compared to other oxide materials proposed as promising photovoltaic absorbers (see Table 1) [20,21,22,23,24,25]. The SLME in Table 1 is evaluated for the thickness *L* = 2.0 μm and temperature *T* = 300 K; the SLME is strongly dependent on the thickness and the temperature of the photovoltaic absorber. The estimation of the SLME directly suggests that delafossite-CZTO could be an efficient photovoltaic absorber. Here, it is worth recalling that CZTO has several additional advantages for real applications, such as element abundance, low cost, low toxicity, and high stability under an ambient environment.

To synthesize the promising quaternary compound CZTO, we have tried high-temperature sintering up to *T* = 1673 K and under ambient conditions using an alumina crucible. Appendix A shows the X-ray diffraction (XRD) 2θ scan of the synthesized specimen. The XRD pattern is unveiled but is not to be analyzed with theoretically predicted CZTO structures. Rather, the XRD pattern can be interpreted with four stable binary and ternary compounds: Zn_2_SnO_4_, SnO_2_, Cu_2_O, and CuO. The XRD peak intensities of Cu_2_O and CuO are relatively weaker than those of Zn_2_SnO_4_ and SnO_2_, which might be attributed to the volatile nature of Cu atoms and the reaction between Cu_x_O and the alumina crucible [26]. This result indicates that the precise control of stoichiometry (particularly the amount of Cu atoms) is crucial for synthesizing CZTO. Still, there are various alternative methods to try for synthesizing CZTO compounds; for example, quartz sealing, pressure-controlled heat treatment, and rapid liquid phase synthesis through arc discharge, as well as thin-film synthesis via physical vapor deposition methods like sputtering and pulsed laser deposition (PLD). The experimental validation of our work remains for future work.

## 3. Conclusions

Finding efficient and cost-effective materials for photovoltaic absorbers is crucial for advancing solar energy technology and achieving sustainable energy solutions. Our study explored CZTO compounds as promising alternatives to conventional absorber materials, employing ab initio DFT calculations and the PSO algorithm. We identified the most stable crystal structure, known as delafossite structure, which exhibited a direct band gap conducive to effective light absorption. SLME analysis underscored CZTO’s high efficiency potential. If the discovered phase of CZTO is synthesized, it will present a compelling option for next-generation photovoltaic cells, with the material’s abundance, low cost, low toxicity, and potential stability under operating conditions.

## Figures and Tables

**Figure 1 nanomaterials-13-03111-f001:**
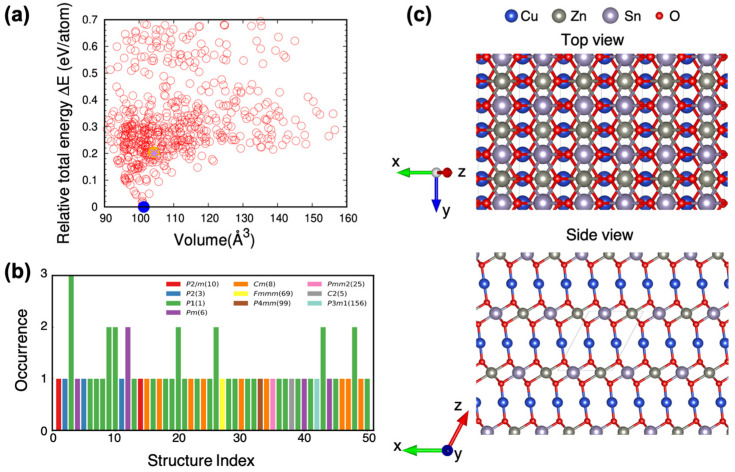
Computational search for thermodynamically stable Cu_2_ZrSnO_4_ (CZTO). (**a**) Relative total energy ΔE with respect to the most stable structure as a function of volume for CZTO. The stable structures are found by using the particle swarm optimization (PSO) algorithm. The most stable structure is color-coded blue. (**b**) The low-energy structures of CZTO are found by the PSO algorithm. The crystal structures are color-coded and ordered according to the energy hierarchy, i.e., the first one is the most stable structure. (**c**) Top and side views of the most stable structure within our materials search in CZTO: delafossite structure. The black solid line shows a unit cell of the structure.

**Figure 2 nanomaterials-13-03111-f002:**
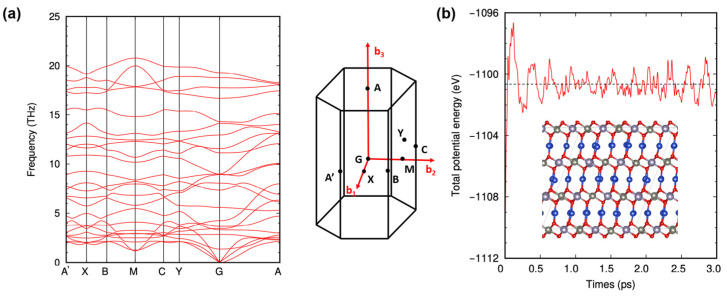
Structural stability of delafossite-CZTO. (**a**) Phonon band structures along the high-symmetry points. (**b**) Total potential energy as a function of time during canonical MD simulations at room temperature with the final structural at the end of the simulation time of 3 ps (inset). The black dashed line represents the potential average from the MD simulation at room temperature.

**Figure 3 nanomaterials-13-03111-f003:**
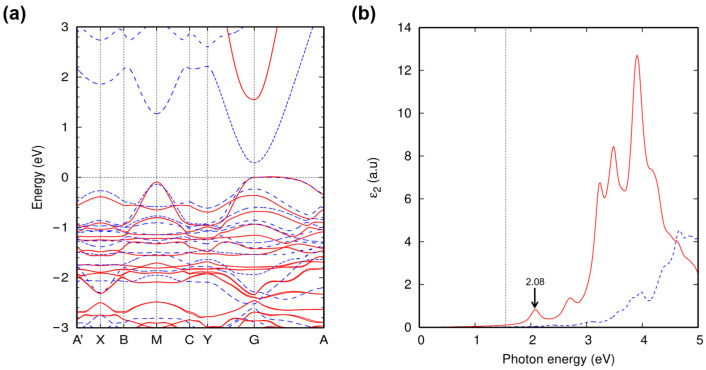
Electronic structure of delafossite-CZTO. (**a**) Band structure of delafossite-CZTO. Blue dashed and red solid lines represent energy bands calculated from DFT-PBE and GW_0_, respectively. The valence band maximum is set to zero. (**b**) Absorption spectra ε_2_ from GW + BSE (red solid line) and GW + RPA (blue dashed line). The dotted gray lines indicate photon energy equal to the quasiparticle energy gap.

**Figure 4 nanomaterials-13-03111-f004:**
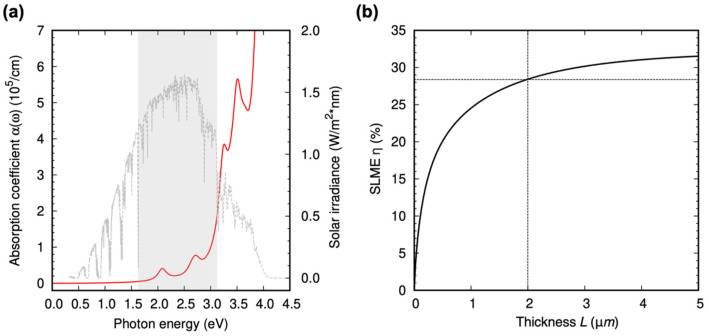
Absorption properties and figure-of-merit of delafossite-CZTO. (**a**) Absorption coefficient from GW + BSE for bulk Cu_2_ZnSnO_4_. The gray box indicates the energy range of visible light. The AM1.5 is shown with gray lines. (**b**) The SLME (*η*) as a function of thickness *L* at 300 K.

**Table 1 nanomaterials-13-03111-t001:** Spectroscopic limited maximum efficiency (SLME) of oxide compounds proposed as promising photovoltaic absorbers. The SLME are evaluated for the thickness *L* = 2.0 μm and temperature *T* = 300 K. The SLME of CdTe is also included for reference [20,21,22,23,24,25].

	SLME (%)		SLME (%)
Cu_2_ZnSnO_4_	28.2	Cu_2_O	0.5
CuGaO_2_	32.6	Cu_2_O:Zn	8.0
CuInO_2_	31.9	Ba_2_SnNbO_6_	26.5
SnO	5.3	Na_2_Tl_0.25_Bi_0.75_O_6_	15.5
2D-SnO	13.4	SrBaVBiO_6_	16.8
Sn_0.75_Zn_0.25_O	22.2	CdTe (ref)	30.1

## Data Availability

The data presented in this study are available on request from the corresponding author.

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
