# Peer review of "Theoretical Investigation of Delafossite-Cu2ZnSnO4 as a Promising Photovoltaic Absorber"

_nanomaterials, 2023, doi:10.3390/nano13243111_

Round 1
Reviewer 1 Report
Comments and Suggestions for Authors
The investigation about Cu2ZnSnO4 in this manuscript proved that Cu2ZnSnO4 would be a promising photovoltaic absorber. I think it can be accepted. It would be better if the authors can present several potential solar cell structures.
Reviewer 2 Report
Comments and Suggestions for Authors
The authors present an interesting work about the simulation of a very novelty absorber material for photovoltaic applications. The only thing I can say about the manuscript is that I miss the division of the paper in introduction, experimental, results and conclusions. I think that more data can be given about the experimental part and also, discuss deeper the results, finishing giving a conclusions. Otherwise, the manuscript is well-written and very clear.
Reviewer 3 Report
Comments and Suggestions for Authors
The authors of this paper mainly conducted a theoretical investigation of delafossite-Cu2ZnSnO4. As we are in need of energy, this type of photovoltaic research attracts many researchers across the globe. This is regular work, and conducted in a routine manner. I have a few concerns before its acceptance.
1. If the authors detect that the cu atoms have a precise control on stoichiometry, then the theoretical study should conduct accordingly. It will be interesting if the authors can give an idea how the concentration of Cu affects the photovoltaic properties. This can lead the paper to an interesting article. Otherwise it is just a normal routine work.
2. What does it mean by indicating the first peak in absorption spectra as 2.08? Please clarify.
3. Why is the SLME measured for Cu2ZnSnO2, not Cu2ZnSnO4?
4. Figures should be placed where it is explained. It is annoying to check by scrolling down. Also, Fig 3 is mentioned as Fig.2 in text.
Comments on the Quality of English LanguageMinor editing required.
Reviewer 4 Report
Comments and Suggestions for Authors
The manuscript presents an interesting topic on examine the potential of another com-17 pound Cu2ZnSnO4 (CZTO) with a similar composition to CZTS as a promising alternative
A few comments and questions are due before we proceed with this submission:
1-please clarify what's the impact of this on absorption profile of the material as you observe the d and p orbitals in the valence and conduction bands in delafossite-CZTO in relation to the band-structure.
2-please clarify why the electron-hole interactions shift the overall absorption spectra to the lower photon energy range in fig. 2b?
3-in the introduction, the citation is essential to: https://doi.org/10.1016/j.renene.2021.09.035
4-please clarify why XRD peak intensities of Cu2O and CuO are relatively weaker than those of Zn2SnO4 and SnO2 in correlation to the table 1.
When authors take my comments in and reply to my questions above and revise the manuscript accordingly, i can reconsider my decision. do not skip any comments above.
Round 2
Reviewer 4 Report
Comments and Suggestions for Authors
my comments properly inserted in new version. i can accept now